# Automatic Differentiation:
# Inverse Accumulation Mode

**Jeffrey Mark Siskind**
School of Electrical and Computer Engineering
Purdue University
West Lafayette, IN 47907-2035
qobi@purdue.edu

## Abstract

We show that, under certain circumstances, it is possible to automatically compute Jacobian-inverse-vector and Jacobian-inverse-transpose-vector products about as efficiently as Jacobian-vector and Jacobian-transpose-vector products. The key insight is to notice that the Jacobian corresponding to the use of one primitive arithmetic operator is of a form whose sparsity is invariant to inversion. This technique has the potential to allow the efficient direct calculation of Newton steps.

Automatic Differentiation (AD) is the mechanical transformation of computer programs to calculate derivatives of interest, with useful complexity guarantees. The two most important "modes" of AD are forward and reverse, which access the Jacobian (the matrix of derivatives of each output of the computation with respect to each input) by multiplication, or transpose-multiplication, with a vector. Here we consider first-order numeric computations, where inputs and outputs are vectors of reals. Given the primal computation $y = f(x)$ with $f : \mathbb{R}^m \to \mathbb{R}^n$ and therefore $x : \mathbb{R}^m$ and $y : \mathbb{R}^n$, we use $\mathbf{J}_{f(x)} : \mathbb{R}^{n \times m}$ for the Jacobian of $f$ at $x$, whose $(i,j)^{\text{th}}$ element is $\partial f_i(x)/\partial x_j$. Forward and Reverse AD compute $\acute{y} = \mathbf{J}_{f(x)}\, \acute{x}$ and $\grave{x} = \mathbf{J}_{f(x)}^\top\, \grave{y}$ respectively. Our objective here is to find an efficient way to solve for the starred vectors in each of

$$\grave{x} = \mathbf{J}_{f(x)}^\top\, \grave{y}^* \qquad\qquad \acute{y} = \mathbf{J}_{f(x)}\, \acute{x}^* \qquad\qquad (1)$$

If this can be done efficiently, it would allow efficient Newton steps (where $f$ is a gradient calculation, say) and other sorts of second-order optimization. For this to be well posed it is necessary for $\mathbf{J}_{f(x)}$ to be invertible, so $n = m$.

Let us review Forward and Reverse AD. Since we are evaluating $f$ at a point $x$, we consider control flow resolved and represent the computation as a data flow graph: a DAG whose edges hold reals and whose vertices represent numeric basis functions. There are $n$ edges entering from the inputs $x_1, \ldots, x_n$, and $n$ exiting to $y_1, \ldots, y_n$. If we topologically sort the data flow graph, and cut it before and after each vertex, we see that the computation proceeds through a sequence of $T + 1$ machine states, $\mathbf{x}_0, \ldots, \mathbf{x}_T$, where the initial and final states are the input and output of the computation, $\mathbf{x}_0 = x$ and $y = \mathbf{x}_T$. We can denote the transition function from one machine state to the next by $\mathbf{x}_t = f_t(\mathbf{x}_{t-1})$ and the Jacobian of $f_t$ at $\mathbf{x}_{t-1}$ by $\mathbf{J}_t$, keeping in mind that $f_t$ involves applying a single numeric basis function to some elements of $\mathbf{x}_{t-1}$ and putting the result in some elements of $\mathbf{x}_t$, copying the other elements unchanged. Since $f = f_T \circ f_{T-1} \circ \cdots \circ f_2 \circ f_1$, the Jacobian matrix is a product, $\mathbf{J}_{f(x)} = \mathbf{J}_T \mathbf{J}_{T-1} \cdots \mathbf{J}_2 \mathbf{J}_1$, and Forward and Reverse AD amount to appropriate associativity

$$\acute{y} = \mathbf{J}_T(\mathbf{J}_{T-1} \cdots (\mathbf{J}_2(\mathbf{J}_1\, \acute{x})) \cdots) \qquad\qquad \grave{x} = \mathbf{J}_1^\top (\mathbf{J}_2^\top \cdots (\mathbf{J}_{T-1}^\top(\mathbf{J}_T^\top\, \grave{y})) \cdots) \qquad (2)$$

Preprint. Under review.

Solving (1) in the form of (2) while assuming each $\mathbf{J}_t$ is invertible is the basic idea of *Forward Inverse Accumulation* and *Reverse Inverse Accumulation:*

$$\acute{y}^* = \mathbf{J}_T^{-\top}(\mathbf{J}_{T-1}^{-\top}\cdots(\mathbf{J}_2^{-\top}(\mathbf{J}_1^{-\top}\acute{x}))\cdots) \qquad \acute{x}^* = \mathbf{J}_1^{-1}(\mathbf{J}_2^{-1}\cdots(\mathbf{J}_{T-1}^{-1}(\mathbf{J}_T^{-1}\acute{y}))\cdots) \quad (3)$$

These will be practical if the matrix-vector products $\mathbf{J}_t^{-1}\acute{y}$ and $\mathbf{J}_t^{-\top}\acute{x}$ can be calculated efficiently. Assuming the computation of $f$ is constant-width, so $\mathbf{x}_t : \mathbb{R}^n$, and invertible, then each $f_t$ must write its result to a slot where one of the inputs to the invoked basis function was stored, yielding Jacobians of the form (for unary and binary basis functions $g$ and $h$)

$$a = \frac{\partial g(\mathbf{x}_{t-1}[R_t])}{\partial \mathbf{x}_{t-1}[R_t]} \qquad\qquad a = \frac{\partial h(\mathbf{x}_{t-1}[R_t], \mathbf{x}_{t-1}[S_t])}{\partial \mathbf{x}_{t-1}[R_t]}$$
$$b = \frac{\partial h(\mathbf{x}_{t-1}[R_t], \mathbf{x}_{t-1}[S_t])}{\partial \mathbf{x}_{t-1}[S_t]}$$

$$(4)$$

We now note that these can be trivially inverted! If we consider only variables involved in the basis function being invoked, and reorder them so the output values are first, a basis function with $k$ inputs and a scalar output results in

$$\mathbf{J}_t = \left(\begin{array}{c|ccc} a & b_1 & \cdots & b_{k-1} \\ \hline \mathbf{0} & & \mathbf{I} & \end{array}\right) \qquad \mathbf{J}_t^{-1} = \left(\begin{array}{c|ccc} \frac{1}{a} & -\frac{b_1}{a} & \cdots & -\frac{b_{k-1}}{a} \\ \hline \mathbf{0} & & \mathbf{I} & \end{array}\right) \qquad (5a)$$

We can generalize from scalar to $l$ outputs, giving the form

$$\mathbf{J}_t = \begin{pmatrix} \mathbf{A} & \mathbf{B} \\ \mathbf{0} & \mathbf{I} \end{pmatrix} \qquad \mathbf{J}_t^{-1} = \begin{pmatrix} \mathbf{A}^{-1} & -\mathbf{A}^{-1}\mathbf{B} \\ \mathbf{0} & \mathbf{I} \end{pmatrix} \qquad (5b)$$

where $\mathbf{A} : l \times l$ and $\mathbf{B} : l \times (k-l)$. Note: $\mathbf{J}_t$ is not structurally symmetric, and $\mathbf{J}_t^{-1}$ has the same structural sparsity as $\mathbf{J}_t$. Although the amount of arithmetic is the same as for conventional Forward and Reverse modes, these are transposed, so Forward Inverse Mode writes to the derivative-related quantities associated with *all* involved variables of each basis function invocation, while Reverse Inverse Mode writes only to the quantities associated with slots *written to* in the primal computation of each basis function. Figure 1 illustrates all four AD modes on a simple program.

In addition to tuning our implementation and seeking interesting constant-width computations to which this might be fruitfully applied, our current work focuses on relaxing the constant-width assumption both locally (by allowing and chunking "lumps" in the primal flow graph) and globally (by using pseudoinverses in place of inverses for computations whose width only grows or shrinks but not both.)

The idea of direct calculation of the solution a linear system involving the Jacobian was introduced by Griewank (1990), and elaborated by Dixon (1991), Utke (1996), and Hossain (1998, Chapter 4), using a framework in which the multiple $\mathbf{J}_t$ matrices here are replaced by a single much larger matrix. Because it is not compositional, that framework seems less amenable to efficient implementation.

This motivates inclusion of four new programming-language primitives to perform each of the four AD modes:

|  | forward | reverse |
|---|---|---|
| noninverted | $\overrightarrow{\mathcal{J}}\, f\, x\, \acute{x} \triangleq \mathbf{J}_{f(x)}\, \acute{x}$ | $\overleftarrow{\mathcal{J}}\, f\, x\, \grave{y} \triangleq \mathbf{J}_{f(x)}^\top\, \grave{y}$ |
| inverted | $\overrightarrow{\mathcal{J}}\, f\, x\, \grave{x} \triangleq \mathbf{J}_{f(x)}^{-\top}\, \grave{x}$ | $\mathcal{L}\, f\, x\, \acute{y} \triangleq \mathbf{J}_{f(x)}^{-1}\, \acute{y}$ |

which obey a variety of algebraic invariants:

$$(\overleftarrow{\mathcal{J}}\, f\, x\, \grave{y}) \cdot \acute{x} = \grave{y} \cdot (\overrightarrow{\mathcal{J}}\, f\, x\, \acute{x}) \qquad \overrightarrow{\mathcal{J}}\, f\, x \circ \mathcal{L}\, f\, x = \mathbf{id} \qquad \overrightarrow{\mathcal{J}}\, f\, x \circ \overleftarrow{\mathcal{J}}\, f\, x = \mathbf{id}$$

$$\grave{x} \cdot (\mathcal{L}\, f\, x\, \acute{y}) = (\overrightarrow{\mathcal{J}}\, f\, x\, \grave{x}) \cdot \acute{y} \qquad \overleftarrow{\mathcal{J}}\, f\, x \circ \overrightarrow{\mathcal{J}}\, f\, x = \mathbf{id} \qquad \mathcal{L}\, f\, x \circ \overrightarrow{\mathcal{J}}\, f\, x = \mathbf{id}$$

$$(6)$$

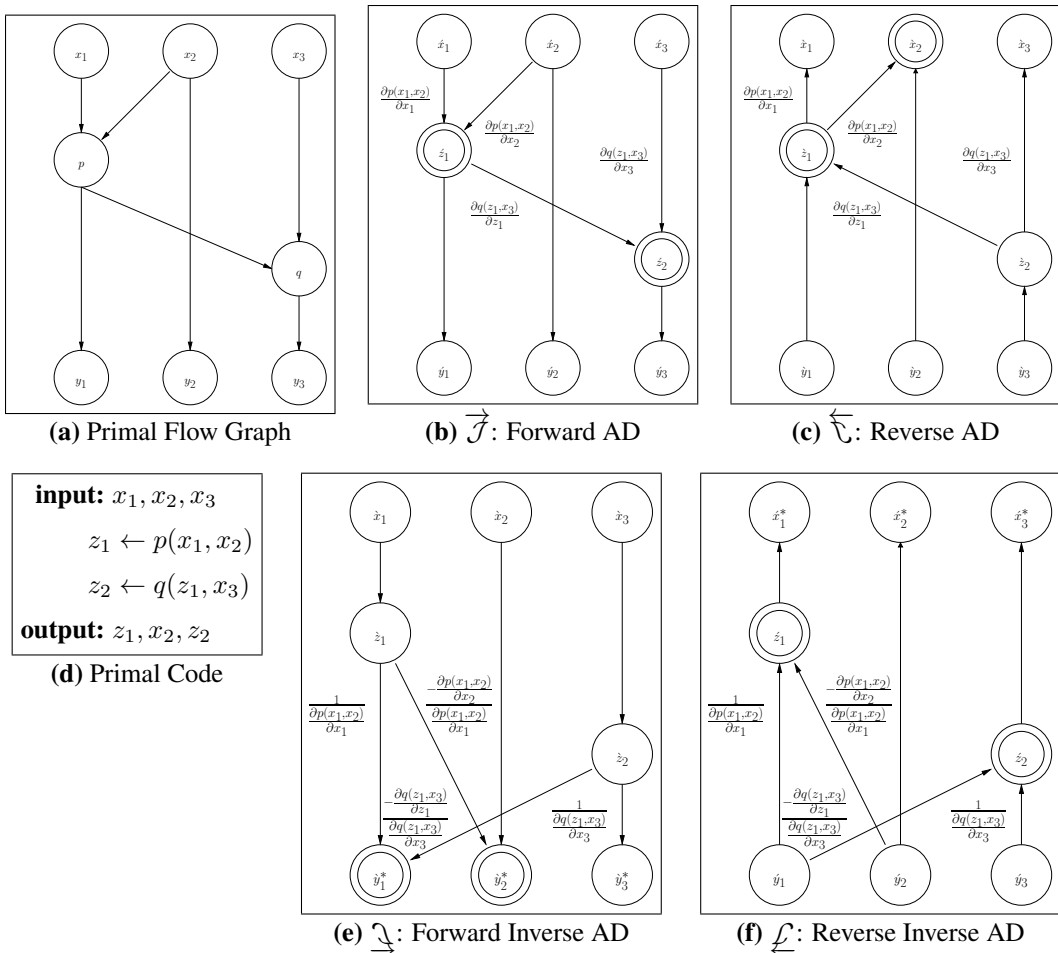

**(a)** Primal Flow Graph

**(b)** $\overrightarrow{\mathcal{J}}$: Forward AD

**(c)** $\overleftarrow{\mathcal{J}}$: Reverse AD

**input:** $x_1, x_2, x_3$

$$z_1 \leftarrow p(x_1, x_2)$$

$$z_2 \leftarrow q(z_1, x_3)$$

**output:** $z_1, x_2, z_2$

**(d)** Primal Code

**(e)** $\underset{\longrightarrow}{\mathcal{J}}$: Forward Inverse AD

**(f)** $\underset{\longleftarrow}{\mathcal{L}}$: Reverse Inverse AD

Figure 1: Illustration of all four AD modes for the straight-line code in (d). This corresponds to the data flow graph (a). The intent is that there are three registers, $r_1$, $r_2$, and $r_3$, illustrated by the three columns in (a) from left to right. These are initialized with $x_1$, $x_2$, and $x_3$ respectively. Since $r_1$ is not used after the first line of code, it is overwritten with $z_1$. Since $r_3$ is not used after the second line of code, it is overwritten with $z_2$. Forward mode and reverse mode are shown in (b) and (c) respectively. In these graphs, addition occurs whenever there is fan in to a vertex (the circled vertices) and labels on edges denote multiplication by the indicated coefficient. Reverse mode is derived from forward mode by edge reversal, which can change which vertices perform addition due to fan in. Forward inverse mode and reverse inverse mode are shown in (e) and (f) respectively. These have the same vertices as forward mode and reverse mode but different edges and edge labels, which changes which vertices perform addition due to fan in. Again, forward inverse mode is derived from reverse inverse mode by edge reversal.

### Acknowledgments

This work is joint with Barak A. Pearlmutter, and was supported, in part, by US National Science Foundation (NSF) grants 1522954-IIS and 1734938-IIS, and by a US Intelligence Advanced Research Projects Activity (IARPA) grant via Department of Interior/Interior Business Center (DOI/IBC), contract number D17PC00341.

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
