# OpenReview forum: "Automatic Differentiation: Inverse Accumulation Mode"
_NeurIPS.cc/2019/Workshop/Program_Transformations — Program Transformations @NeurIPS2019 Poster_

### Official Review · AnonReviewer1 · 2019-09-26
**A way to compute inverse-transposed-Jacobian times vector efficiently, potentially extremely efficient, albeit with stringent limitations.**

**Confidence:** 4
**Rating:** 7

**Review:**

A good abstract, clear and well written, on a potentially very efficient way to compute products of the *inverse* Jacobian (transposed or not) with a vector.
Transposed Jacobian times vector (i.e. adjoint mode of AD, or backpropagation of ML) is a well-known nightmare because transposition implies reversed execution order and therefore the hassle of storing intermediate values or recomputing them many times. In some cases, the desired result is rather the Inverse Transposed Jacobian times vector. This additional inversion implies a additional reversal of the execution order, i.e. in the end no reversal at all, which would eliminate the storing/recomputing hassle.
Unfortunately this requires to choose the individual steps of the computation so that their elementary Jacobian is invertible, and this is not granted in general (the abstract maybe skims a little too fast on this issue, i.e. on line 5 of page 2). Readers may want to hear more about the best granularity of these "invertible-Jacobian" steps (suggestions?).
The underlying idea of the abstract has been published in the past by several authors, as the abstract rightly points out. The abstract has the merit to reformulate it in terms of special operators that could be first-class primitives of an differentiation-extended programming language.

---

### Official Review · AnonReviewer2 · 2019-09-30
**Interesting preliminary work on a method for inverse-Jacobian products**

**Confidence:** 4
**Rating:** 7

**Review:**

The article reviews the problem of automatically computing products between a vector and the inverse of the Jacobian matrix (transposed or not) of an invertible numerical computation, and proposes a formulatoin similar to regular AD.
The introduction of two more compositional operators is interesting.
Most often, writes are considered more expensive than reads, and the primal tends to minimize them. Reverse Inverse AD, with the same write patterns, would then be quite useful.
I'm uncertain how widely this transformation can be applied, as the model needs to be reversible, but it could be useful for RevNet or flow-based (reversible) generative models.

---

### Public Comment · ~Andreas_Griewank2 · 2019-10-02
**partial "a" may not be too small**

Of course we need not only mathematical reversibility but some numerical stability.
Also we need the final state of the whole memory space to recover its original state.
Many times there is a projection at the end which only picks out some values as
dependent variables. The other stuff can be stored of course, so yes good idea.

---

### Decision · Program_Chairs · 2019-10-01

**Decision:**

Accept (Poster)

**Comment:**

The reviewers pointed out that although the proposed technique has potential, its stringent limitations and a lack of proposed applications make it arguably too preliminary for an oral presentation.